# Requirements engineering issues causing software development outsourcing failure

**Javed Iqbal**[1]*, **Rodina B. Ahmad**[2], **Muzafar Khan**[3], **Fazal-e-Amin**[4], **Sultan Alyahya**[4], **Mohd Hairul Nizam Nasir**[2], **Adnan Akhunzada**[1], **Muhammad Shoaib**[4]

**1** Department of Computer Science, COMSATS University, Islamabad, Pakistan, **2** Faculty of Computer Science and Information Technology, University of Malaya, Kuala Lumpur, Malaysia, **3** Department of Engineering, National University of Modern Languages, Islamabad, Pakistan, **4** College of Computer and Information Sciences, King Saud University, Riyadh, Saudi Arabia

* javediqbal@comsats.edu.pk

**Data Availability Statement:** All relevant data are within the paper and its Supporting Information files.

## Abstract

Software development outsourcing is becoming more and more famous because of the advantages like cost abatement, process enhancement, and coping with the scarcity of needed resources. Studies confirm that unfortunately a large proportion of the software development outsourcing projects fails to realize anticipated benefits. Investigations into the failures of such projects divulge that in several cases software development outsourcing projects are failed because of the issues that are associated with requirements engineering process. The objective of this study is the identification and the ranking of the commonly occurring issues of the requirements engineering process in the case of software development outsourcing. For this purpose, contemporary literature has been assessed rigorously, issues faced by practitioners have been identified and three questionnaire surveys have been organized by involving experienced software development outsourcing practitioners. The Delphi technique, cut-off value method and 50% rule have also been employed. The study explores 150 issues (129 issues from literature and 21 from industry) of requirements engineering process for software development outsourcing, groups the 150 issues into 7 identified categories and then extricates 43 customarily or commonly arising issues from the 150 issues. Founded on 'frequency of occurrence' the 43 customarily arising issues have been ranked with respect to respective categories (category-wise ranking) and with respect to all the categories (overall ranking). Categories of the customarily arising issues have also been ranked. The issues' identification and ranking contribute to design proactive software project management plan for dealing with software development outsourcing failures and attaining conjectured benefits of the software development outsourcing.

## 1. Introduction

During information technology outsourcing some or all the IT-related functions are transferred to extrinsic supplier(s) according to a contract [1]. A category of information technology outsourcing is Software Development Outsourcing (SDO) that involves contracting out

**Funding:** The authors extend their appreciation to the Deanship of Scientific Research at King Saud University for funding this work through research group no.RG-1441-490. The funders had no role in study design, data collection and analysis, decision to publish, or preparation of the manuscript.

**Competing interests:** The authors have declared that no competing interests exist.

some or all the software development-related tasks to the vendor(s) [2–3]. The concept of SDO is gaining popularity swiftly [4] as it proclaims the benefits of both parties [5]. European firms contract out software development to countries like India, Vietnam and China [6].

There are two main classes of the reasons for outsourcing [7–9]: I. Advantages of outsourcing for example cost savings, exploiting superior technologies and capabilities, and utilizing inner resources optimally, ii. Organizations' restrictions, for example, poor management and scarceness of the apposite resources. The vendor is profited by the enrichment of expertise and by learning how clients' requirements can be satisfied [10]. Thus, vendor is capable of adding significant value to clients' supply chains [11]. SDO has several types [12–13] like onshoring [14–15], nearshoring [14], offshoring [14], distributed software development [16–17] and Global Software Development (GSD) [16–18].

The projects are outsourced for software development to attain predicted advantages, but several jeopardies are associated with SDO [10]. Rate of failure is high in case of such projects, for example, 40% of the offshored projects did not achieve foreseen advantages [19]. The rate of failure in case of GSD is 50% [6, 13]. Surveys prove that success rate in case of SDO is only 50% [20]. The issues that are originated from Requirements Engineering (RE) process, are one of the main reasons of SDO failure [6, 8, 21–22].

RE is the most crucial activity during Software Development Life Cycle (SDLC) which also affects other SDLC activities substantially [23–24]. A study shows that RE related errors occur frequently during SDLC [25]. According to an industrial survey of the RE problems confronted by 12 software development companies, RE related errors are 48% of the total number of SDLC errors [24]. These problems are augmented manifold in the case of SDO because of the physical dispersion of stakeholders [18, 26–27]. Thus, many issues are created for RE process in the case of SDO [18, 28]. Therefore, customarily occurring or arising issues of the RE process for SDO must be identified and ranked to design a proactive strategy for addressing SDO failure and hence attaining the benefits of SDO. While finding common issues of the RE process for SDO, the categories of such issues should also be known so that the issues could be grouped into the corresponding categories.

In this context, this study frames the following Research Questions (RQs):

RQ1: Which are categories of the issues of the RE process for SDO?

RQ2: Which are customarily or frequently arising issues of the RE Process for SDO?

Along with the identification of the common SDO RE process issues, the issues need to be ranked

to plan a proactive and workable strategy for addressing the issues. This leads to the third RQ:

RQ3: What is the ranking of each:

3.1. Customarily arising issue of the RE process for SDO with regards to the respective category

of the issue (Category-wise ranking)?

3.2. Customarily arising issue of the RE process for SDO with regards to issues belonging to all

the categories (Overall ranking)?

3.3. Category of the issues of RE process for SDO?

This paper is organized as follows: section 2 highlights the related work, section 3 expresses the research methodology adopted for this research work whereas section 4 describes results. Section 5 presents discussions and section 6 is regarding limitations of the study. Finally, section 7 concludes the paper and specifies future directions.

## 2. Related work

Several studies in the current literature focus on the SDO RE process issues. In the study [29], the prime focus is on 'requirements understanding' in GSD. According to [30], the distributed software development stresses on thorough understanding of the RE related activities which require collective attempts from the dispersed stakeholders. A framework called PBURC has been presented and tested to collect and validate data during the RE process that involves varied backgrounds and services [31]. The usage of MAS (Multi Agent System) architecture has been described to lessen the problems of distributed RE process particularly for verification and validation [32]. To comprehend the convolutions of the GSD RE process, functioning of twenty-four virtual teams has been analyzed during the requirements' definition [23]. Through a field study, D. Damian has investigated the impact of the geographically distributed stakeholders on the RE process [33]. Depending on the exposure of RE related tasks and the GSD problems, several GSD RE models have been presented and assessed in [34]. The V model has been recommended to extract and choose the requirements for a product release in the case of dispersed stakeholders [35]. The knowledge distribution and reuse in the case of global RE has been debated in [36]. To address the challenge of the huge numbers of distributed end users, a unified online approach has been introduced in [37].

Damian et al. [38] highlight the significance of human coordinator for an effective distributed RE process. From the point of view of a software developer, the consequences of following a poor RE process, in the case of software development project outsourcing, have been explored in [39]. Another field study reveals certain inferences regarding the GSD RE process [40]. RE related activities create project management challenges in the case of GSD and the factors that cause GSD project failure are mostly associated with requirements [41]. Because of the inappropriate 'understanding of requirements', vendors are unable to apply technical skills [42]. Misunderstanding of requirements is a challenge in the case of GSD projects and to manage GSD projects successfully all the requirements are needed to be satisfied [43]. The requirements stability is one of the crucial factors that affect decisions about the task allocation during GSD projects [44]. Effective coordination among virtual team members becomes difficult because of changing requirements, therefore, unstable requirements hamper virtual software development teams' operations [45]. Requirements elicitation and documentation is a challenge in the case of GSD [46]. The issues like insufficient understanding of the requirements, inappropriate requirements change management and quickly changing requirements lead to integration failures in the case of GSD [47]. The methods that are employed to specify and validate the requirements for collocated development of software, are not effective in the case of GSD. The study [48] advocates a method to document and validate the requirements in the case of GSD. To apply the method, requirements graph and validation matrices are generated. To address the RE process issues that occur because of the physical dispersion of stakeholders in the case of GSD, a RE process has been proposed especially for GSD that is based on lexicon model and scenarios [49]. The significance of project management for RE and requirements change management in the case of GSD, has been explored in [50]. For this purpose, two frameworks have been proposed and validated through survey and interviewing. To facilitate the requirements change management in the case of GSD, a three stage method has been proposed: i. Changes' understanding, ii. Change Analysis, and iii. Changes' finalization [51]. Geographical, cultural and temporal distances cause communication risks during the requirements change management in the case of GSD. To address such communication risks, a framework has been proposed [52].

Thus, numerous studies in the contemporary literature focus on the issues of RE process for SDO but no study presents commonly or frequently arising SDO RE process issues.

Besides, several SDO RE issues are encountered by SDO practitioners but have not been reported in the literature. This research work intends to present a comprehensive list of the SDO RE process issues based on SDO RE process issues identified from the current literature as well as from the SDO industry. To address the SDO failures and hence to attain the benefits of SDO, the research work extracts the commonly occurring SDO RE process issues and also ranks such issues. The following section clarifies the research methodology adopted to carry out the research work.

## 3. Research methodology

The research work for this study, being part of PhD work, has been approved by the Candidature Defense Committee. The questionnaire surveys are only human related subject of this study. Before conducting the surveys, the verbal consent has been obtained from the potential participants or from their respective organizations. No personal data has been presented or analyzed in any form in this study. The responses have been presented in an accumulative manner. In this way, privacy and anonymity of the individuals and organizations have been fully protected.

This research work is intended to identify the customarily arising issues of the RE process in the case of SDO. Therefore, as the step I, categories of the issues have been originated. To find the customarily arising issues of the SDO RE process, initially a comprehensive list of the issues must be organised based on the contemporary relevant literature and industrial perspective. Therefore, step II is to investigate the current literature to find which SDO RE process issues have been presented in the literature. Incorporating the industrial viewpoint is essential to result-oriented and beneficial research. Hence, step II also includes digging out SDO RE process issues that are confronted by the SDO professionals. After exploring the current relevant literature and consulting SDO practitioners, a consolidated list of the SDO RE process issues has been organised in step II. To deal with the SDO RE process issues, the ranking of these issues is crucial based on the 'frequency of issues'. This constitutes the step III of this research work. To design a proactive and doable strategy, the commonly arising SDO RE process issues must be identified. Thus, the ranked list of issues needs to be filtered out to find out frequently arising or common issues. This guides to perform step IV. Thus, to achieve the research objective and answer the RQs, four steps have been executed:

Step I: To categorize the SDO RE process issues, $1^{st}$ questionnaire survey has been conducted by involving the professionals from SDO industry.

Step II: To identify the literature based SDO RE process issues, a thorough literature assessment has been carried out. To explore the additional SDO RE process issues (issues faced by SDO practitioners but not reported in literature), $2^{nd}$ questionnaire survey has been conducted by involving the professionals from SDO industry. Thus, a consolidated list of SDO RE process issues has been prepared.

Step III: To rank the SDO RE process issues, by using the Delphi technique and based on the 'frequency of occurrence' of the issues, $3^{rd}$ questionnaire survey has been conducted by involving the professionals from SDO industry.

Step IV: To extract the customarily arising or common SDO RE process issues from the ranked list of the issues, the cut-off value method has been employed. The customarily arising issues have been ranked within the respective categories and with respect to issues of all the categories. The issues' categories have also been ranked. The overall research methodology has been shown in the Fig 1.

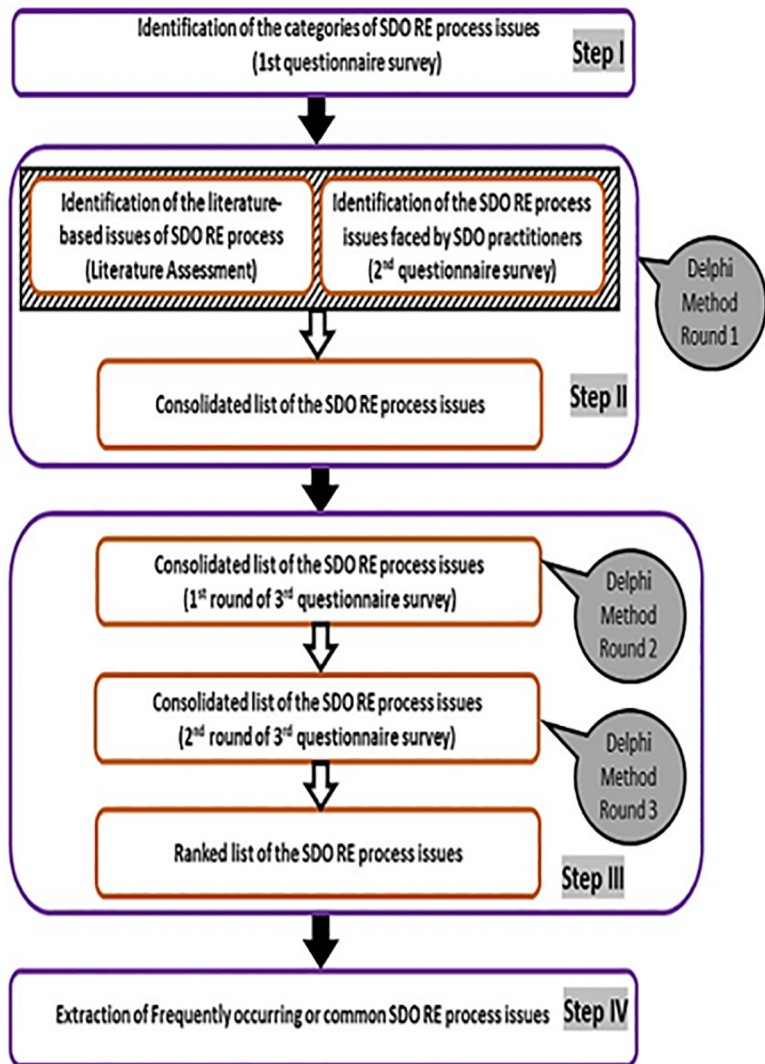

**Fig 1. Steps to identify and rank common SDO RE process issues.** Thus, to identify and rank the customarily arising SDO RE process issues, the relevant literature has been investigated thoroughly and three questionnaire surveys have also been carried out by involving experienced professionals from the SDO industry. The Delphi technique, cut-off value method and 50% rule have also been applied.

### 3.1. Literature assessment

The aim of literature assessment is identification, analysis and interpretation of the current literature pertaining to the certain research question(s) or matter or area of concern [53]. The literature assessment is accomplished through a clear-cut approach that guarantees the comprehensive, impartial and repeatable research process [53]. This research work follows the approach recommended by Kitchenham and Charts [53].

**3.1.1. Data sources for literature assessment.** To search the appropriate studies, five electronic databases have been accessed: i. IEEE Xplore, ii. ACM, iii. Science Direct, iv. Springer Link, and v. Web of Science. Based on key terms, a fundamental search string has been formed and exploited to search the appropriate studies from the various electronic databases.

**3.1.2. Assortment of studies.** The details of the procedure employed for assortment of the relevant studies, have been provided in the Fig 2.

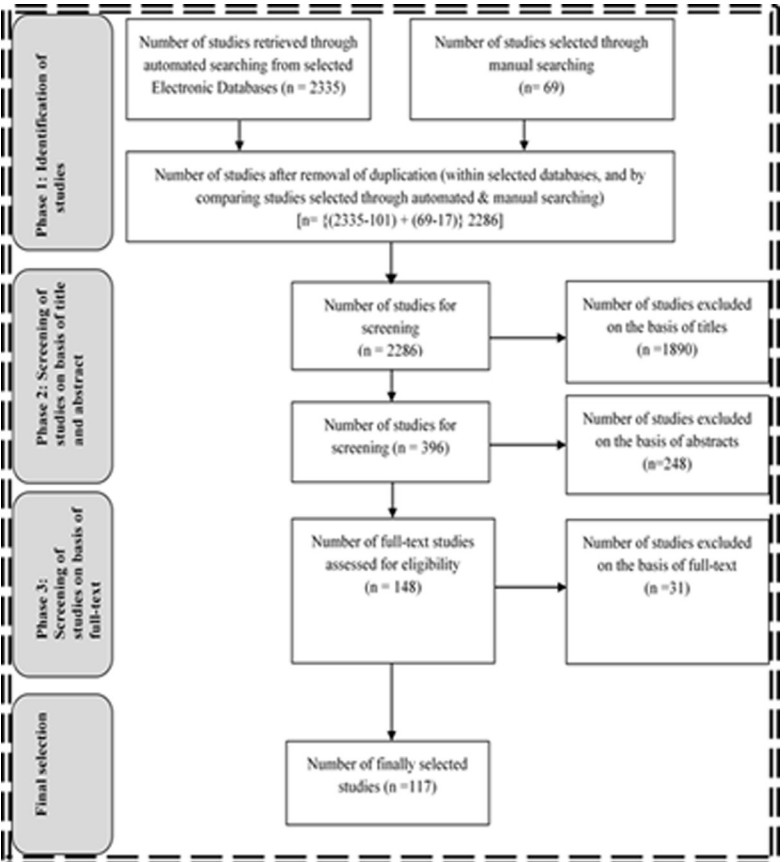

**Fig 2. Studies assortment procedure.**

Thus, after following a laborious search and review process, the 117 studies have been chosen. Out of the 117 assortments, the 77 have been chosen through automatic search whereas the 40 through manual search. Table 1 shows database-wise number of the studies retrieved and then finally selected through automatic search. The details about literature assessment have been provided as **S1 File**.

## 3.2. Questionnaire surveys

Three kinds of the questionnaires that are utilized for survey research are: I. Personally administered, ii. Mailed, and iii. Web-Based [54–55]. Usually the questionnaires employed for survey research contain the questions that are either open-ended or closed-ended [55].

All the questionnaire surveys to carry out this research work, have been performed through semi-supervised approach [56] which has been followed during head-on meetings or by using Computer-Assisted Telephone Interviewing technique [57]. The drop-off/pick-up method has been adopted for distribution and collection of the survey questionnaires [58]. In this method, questionnaires are delivered to the respondents or their representatives and are picked up latter on the mutually decided time. For the drop-off/pick-up method, percentage of the survey participants for filling and returning the questionnaires is quite high [59].

For conducting each survey, a pilot study has been organized [60]. To attain a valid sample of population, Convenience sampling method has been adopted. The survey participants are SDO professionals with designations like project managers, manager operations, senior

managers, quality assurance managers, software engineers, team leads, requirements engineers, analysts, programmers and designers having the minimum experience of five years. The details about survey participants have been provided as **S2 File**.

### 3.3. Employing the delphi technique

This study employs the Delphi technique to find and rank the customarily arising issues of the SDO RE process. The Delphi technique involves a repetitive process that comprises of two or three or more number of cycles. A cluster of experts, in a specific area, participates in each cycle and every expert gives his/her opinion. After completion of each cycle, both the accumulative result of that cycle and an expert's individual response are provided to every expert. Then every expert is requested to reassess his/her individual opinion keeping in view overall result, and so on [61–63]. The Delphi technique is adopted to grow the unanimity among experts or to congregate the judgment of experts on the certain issue(s) [61–63].

### 3.4. The 50% rule

The 50% rule means if at least 50% respondents are in the support of an opinion then that opinion is accepted. For several studies similar rule has been followed [64–66].

### 3.5. The cut-off value method

In the cut-off value method, certain items or factors are selected or dropped based on a cut-off value [67]. In this study, the cut-off value has been decided in the two ways: i. By calculating the average of 'highest mean' and 'lowest mean'. ii. By calculating average of 'all means'.

## 4. Results

The study presents results and discussions with respect to the various steps that have been presented in the research methodology section, and have been carried out to identify and rank the commonly occurring SDO RE process issues along with the issues' categories.

### 4.1. Identifying categories of the SDO RE process issues (step I)

This study employs a questionnaire survey to finalize the categories for the issues of the SDO RE process by following the guidelines presented by Kitchenham and Pfleeger [68]. The questionnaire contains nine potential categories of the issues, extracted from literature, 'Yes' or 'No' options to select or drop a potential category and option for mentioning any other category for the issues, if not specified in the given list of potential categories. Out of the 200 distributed questionnaires, 115 have been received back and 105 have been chosen for the data analysis based on the quality criteria.

   **4.1.1. Criterion for the identification of issues' categories.**   The 50% rule has been applied to determine the categories of the issues. Out of the 9 potential categories, for 7 categories, at least 50% participants have opted for the option of 'Yes'. For the remaining 2 categories, percentage of 'Yes' option is below 50%. Table 2 presents the results.

   Fig 3 portrays the results.

   **4.1.2. Categories of the SDO RE process issues.**   Suppose $CAT_1$, $CAT_2$, ..., $CAT_7$ represent sets of issues belonging to the communication, knowledge management and awareness, cultural diversities, management and coordination, processes and tools, relationship among stakeholders, and requirements centric categories respectively. Then the seven identified categories of the SDO RE process issues are:

i.  Communication issues ($CAT_1$)

**Table 1. Database-wise no. of studies retrieved and finally selected through automatic search.**

| Database | No. of initially retrieved studies | No. of finally selected studies | Percentages of finally selected studies w.r.t. | |
|---|---|---|---|---|
| | | | No. of initially retrieved studies | Total no. of finally selected studies |
| IEEE Xplore | 431 | 39 | 9.04% | 50.65≈51% |
| ACM | 310 | 10 | 3.23% | 12.99≈13% |
| Science Direct | 679 | 08 | 1.18% | 10.39≈10% |
| Springer Link | 662 | 12 | 1.81% | 15.58≈16% |
| Web of Science | 253 | 08 | 3.16% | 10.39≈10% |
| Total | 2335 | 77 | | 100% |

 ii. Knowledge management and awareness issues ($CAT_2$)

 iii. Cultural diversities issues ($CAT_3$)

 iv. Management and coordination issues ($CAT_4$)

 v. Processes and tools issues ($CAT_5$)

 vi. Relationship among stakeholders' issues ($CAT_6$)

 vii. Requirements centric issues ($CAT_7$)

 This helps to answer RQ1

## 4.2. Identifying literature-based and additional issues of the SDO RE process to prepare a consolidated list of the SDO RE process issues (step II)

This study identifies the SDO RE process issues via a rigorous literature assessment and a questionnaire survey (2nd questionnaire survey) by involving professionals from the SDO industry. Two independent investigators have been involved to consolidate and to finalize the issues' list. Ambiguities and anomalies about the expressions or terms, used to describe issues, have been eliminated. The matching issues from the industry and the contemporary literature have been merged. Thus, a consolidated list of the 150 SDO RE process issues has been organized. Out of the 150 issues, the 129 issues belong to literature whereas the professionals from the SDO industry have mentioned the 21 additional issues. Among the 129 issues identified from the literature, 21 issues are associated with 'communication', 21 with 'knowledge management & awareness', 19 with 'cultural diversities', 19 with 'management & coordination', 16 with 'processes & tools', 14 with 'relationship among stakeholders' whereas 19 issues are 'requirements centric'.

 To obtain the additional issues of RE process for SDO, 2nd questionnaire survey has been performed with the SDO practitioners. Instructions given in the study [68] have been followed to carry out this survey. By harnessing the drop-off/pick-up method, the questionnaires have been delivered to 200 SDO industry professionals. The questionnaire contains two portions. The first portion is to accumulate demographic information regarding the participants and second portion is meant for collecting the SDO RE process issues.

 The category-wise literature-based list of the issues has been supplied to the professionals from the SDO industry. The practitioners have been requested that if they believe that any issue in the list must be allocated other category than the present one then they can alter the issue's category by describing the reason for the alteration. The survey participants have also been solicited to state such SDO RE process issues which they have been confronting in the course of their SDO career or regarding which they believe that these issues can occur, but they are not present in the given category-wise literature-based list of the issues.

**Table 2. Results of the 1<sup>st</sup> questionnaire survey to identify the categories of the issues of SDO RE process.**

| Sr. # | Potential categories of issues | Respondents | | | |
|---|---|---|---|---|---|
| | | 'Yes' option | | 'No' option | |
| | | Number | Percentage | Number | Percentage |
| 1 | Communication | 105 | 100% | 0 | 0% |
| 2 | Knowledge management and awareness | 98 | 93.33% | 7 | 6.67% |
| 3 | Cultural diversities | 70 | 66.67% | 35 | 33.33% |
| 4 | Trust | 50 | 47.62% | 55 | 52.38% |
| 5 | Management and coordination | 90 | 85.71% | 15 | 14.29% |
| 6 | Organizational structure | 34 | 32.38% | 71 | 67.62% |
| 7 | Processes and tools | 85 | 80.95% | 20 | 19.05% |
| 8 | Relationship among stakeholders | 80 | 76.19% | 25 | 23.81% |
| 9 | Requirements centric | 100 | 95.24% | 5 | 4.76% |

Totally 110 questionnaires have been returned back. Out of the 110 responses, the 106 responses have been selected for the data analysis depending upon the relevancy of job, experience and company. The SDO industry professionals have stated the 21 additional issues. Among the 21 issues, one belongs to 'communication' category, three belong to 'knowledge management and awareness', three belong to 'cultural diversities', three belong to 'management and coordination', three belong to 'processes and tools', eight issues are 'requirements centric' whereas no additional issue has been reported regarding 'relationship among stakeholders' category.

Thus, by combining the literature-based and the additional issues, we have 22(21+1) communication issues, 24(21+3) knowledge management and awareness issues, 22(19+3) cultural diversities issues, 22(19+3) management and coordination issues, 19(16+3) processes and tools issues, 14(14+0) relationship among stakeholders' issues and 27(19+8) requirements centric issues.

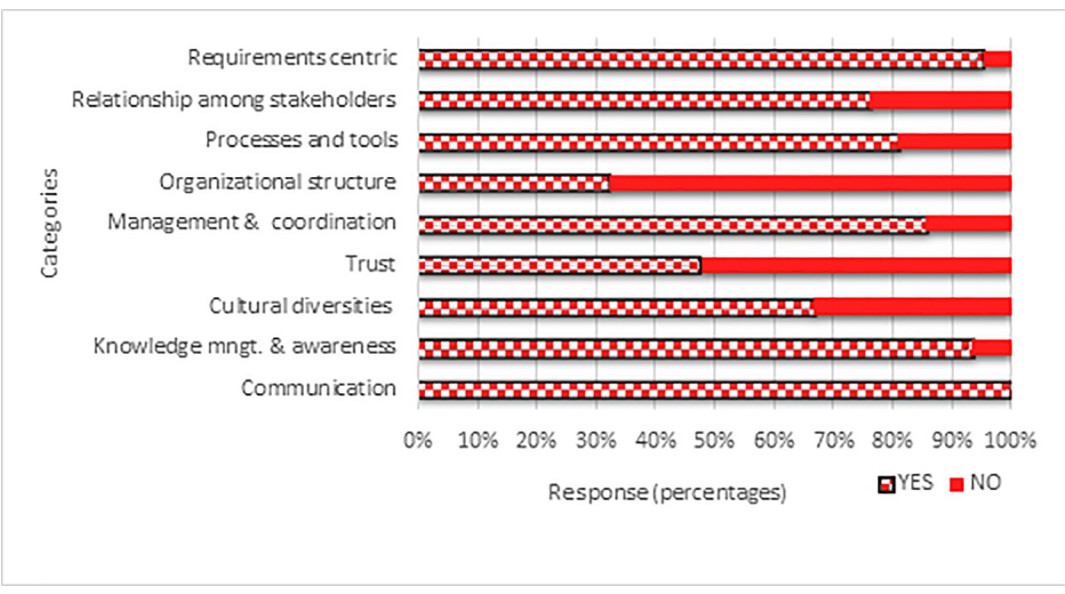

**Fig 3. Percentages of the responses in case of potential categories of SDO RE process issues.**

The issues related to 'communication' are represented as Iss1, Iss2, . . ., Iss22. Likewise, issues associated with 'knowledge management and awareness' are represented as Iss23, Iss24, . . ., Iss46. The 'cultural diversities' issues are symbolized as Iss47, Iss48, . . ., Iss68. Moreover, the issues linked with 'management and coordination' are denoted by Iss69, Iss70, . . ., Iss90. The issues regarding 'processes and tools' are denoted by Iss91, Iss92, . . ., Iss109 and issues related to 'relationship among stakeholders' are denoted by Iss110, Iss111, . . ., Iss123 whereas symbols to represent 'requirements centric' issues are Iss124, Iss125, . . ., and Iss150. **S1 Appendix** presents the 150 issues. Suppose C represents set of the 150 issues belonging to the seven categories then

$C = \{IssX: X \in N \wedge 1 \leq X \leq 150 \}$

So, we can write

$\{Iss1, Iss2, . . ., Iss22\} = CAT_1$

$\{Iss23, Iss24, . . ., Iss46\} = CAT_2$

$\{Iss47, Iss48, . . ., Iss68\} = CAT_3$

$\{Iss69, Iss70, . . ., Iss90\} = CAT_4$

$\{Iss91, Iss92, . . ., Iss109\} = CAT_5$

$\{Iss110, Iss111, . . ., Iss123\} = CAT_6$

$\{Iss124, Iss125, . . ., Iss150\} = CAT_7$

and

$$\bigcup_{i=1}^{7}(CAT_i) = C$$

$$\bigcap_{i=1}^{7}(CAT_i) = \phi$$

## 4.3. Ranking the SDO RE process issues (step III)

The Delphi technique has been employed to rank the issues of SDO RE process.

**4.3.1. The delphi technique.** This study has employed three rounds of the Delphi technique as suggested by preceding studies [62, 69–70]. As far as the number of rounds is concerned, numerous variations of the Delphi technique are pursued. As per recommendations of one study, three rounds are sufficient [71]. The Delphi technique can be curbed to two or three rounds for accomplishing research targets as indicated by several other studies [61, 63, 72–73].

To achieve the objective of this study, three rounds of the Delphi technique have been completed. Similar to the 1st round of preceding studies [62, 72], this study identifies SDO RE process issues at the earlier stage of this research work (see Results, section 4.2). This stage plays the role of the 1st round. The list of the obtained issues has been consolidated as advised in [62, 69–70]. This consolidated list of the 150 issues (provided as **S1 Appendix**) has been utilized while performing 2nd and 3rd rounds. The customarily arising issues could have been extricated after the accomplishment of the 2nd round but to cultivate more accord among the participants, the study carries forward all the issues to the 3rd round. After the completion of the 3rd round, customarily arising issues have been extracted and ranked.

**4.3.2. Performing 2nd and 3rd rounds of the delphi technique.** For the execution of 2nd and 3rd rounds of the Delphi technique, this study organizes two rounds of the questionnaire survey (3rd questionnaire survey). For designing and performing the survey, the study employs procedure presented by Kitchenham and Pfleeger [68]. Before commencing the study, 200 relevant professionals have been pinpointed but only 118 professionals have indicated their

eagerness to take part in the 2$^{nd}$ and 3$^{rd}$ rounds. However, just the 106 professionals have been able to successfully finish both rounds of the study. Several Delphi surveys embroil 100 or additional professionals [74–75].

**4.3.3. Second round.** Amid the 2$^{nd}$ round of the study, a category-wise consolidated list of 150 issues has been delivered to the professionals. The professionals have been invited to allude the 'frequency of occurrence or arising' for every issue. To serve the purpose, a 5- point Likert scale has been utilized as recommended by preceding studies [76–77]. These studies have employed five categories of the issues with respect to occurrence of issues. The categories are:

i.   Almost always (5): The issue is deemed to occur or arise 'Almost always' if it arises nearly each time (means 90% to 100% times).

ii.  Frequently (4): The issue is deemed to arise 'Frequently' if it arises oftenly (means 60% to 89% times).

iii. About half of the time (3): The issue is deemed to arise 'About half of the time' if it arises nearly half the time (means 40% to 59% times).

iv.  Occasionally (2): The issue is deemed to arise 'Occasionally' if it arises less oftenly (means 10% to 39% times).

v.   Rarely (1): The issue is deemed to arise 'Rarely' if it arises hardly ever or never.

The survey has been disseminated to the 118 professionals by utilizing the drop-off/pick-up method. From the 118 surveys, 110 have been collected back. After the completion of 2$^{nd}$ round, the average frequency and the standard deviation have been computed for every issue.

**4.3.4. Third round.** In the 3$^{rd}$ round, surveys have been delivered to only those 110 professionals, by utilizing the drop-off/pick-up method, who reacted amid the 2$^{nd}$ round effectively. Every professional has been equipped with his/her individual 2$^{nd}$ round frequency and also corresponding average frequency for every issue. Every professional has been invited for reconsidering his /her own frequency, for every issue, based on the 2$^{nd}$ round average frequency for that particular issue. Amid the 3$^{rd}$ round, 106 surveys have been collected back. Based on quality criteria, from 106 surveys, 103 have been selected for analyzing data. At the end of the 3$^{rd}$ round, the average frequency and the standard deviation have been computed once again for every issue.

**4.3.5. Results of delphi survey.** S2 Appendix presents the average frequency and the related standard deviation, for each issue, computed for the 2$^{nd}$ and 3$^{rd}$ rounds. This is evident from the S2 Appendix that the average of all the standard deviations computed for 2$^{nd}$ round is 0.729 (Please refer to last row of S2 Appendix). Likewise, the average of all the standard deviations computed for 3$^{rd}$ round is 0.688 (Please refer to last row of S2 Appendix). This illustrates that the standard deviation has lessened after the 3$^{rd}$ round and the consensus among the professionals has improved. The study was concluded after the completion of the 3$^{rd}$ round and following the approach employed during the research work [69].

**4.3.6. Measurement of internal consistency.** After the completion of the 3$^{rd}$ round of the Delphi technique, Reliability Analysis has been performed to measure the internal consistency of the scale. The value of Cronbach Alpha in this case is 0.964. Table 3 presents the value. According to recommendations given in [78–79], the value of Cronbach Alpha equivalent to

**Table 3. Reliability statistics.**

| Cronbach's Alpha | No. of Items |
| --- | --- |
| 0.964 | 150 |

0.7 or greater is 'acceptable', more than 0.8 is deemed 'good' and more than 0.9 implies 'excellent' internal consistency.

**4.3.7. Ranked list of SDO RE process issues.** By capitalizing on the details given in **S2 Appendix**, Table 4 presents means of the response values, for all the 150 issues, in descending order after completion of the 3$^{rd}$ round. Hence Table 4 provides the ranked list of all the 150 SDO RE process issues. Sr. # column also presents ranks of the issues.

## 4.4. Extracting the customarily arising SDO RE process issues (step IV)

To extract the customarily arising SDO RE process issues, from the ranked list of issues, the cut-off value method has been employed.

**4.1.1. Cut-off value method for extracting ccustomarily arising issues.** The technique for the filtration of data items is widely applied in numerous disciplines like psychology, telecommunication and education, and is commonly used to analyze the self-reported studies [67, 80]. This study employs a method analogous to [67].

Utilizing mean values from Table 4,

*The Highest Mean Value (HMV) i.e. for Iss7 = 4.213592*

*The Lowest Mean Value (LMV) i.e. for Iss122 = 1.475728*

Average of HMV and LMV = 2.84466

The cut-off value can be determined from the average of HMV and LMV. This value establishes that issues having means equal to or greater than 2.84466, can be chosen as the customarily arising issues of the SDO RE process.

Based on the average of HMV and LMV, the first 43 issues, presented in Table 4, can be selected as the customarily arising SDO RE process issues. The 43 issues, chosen as the commonly or customarily arising issues are:

Iss1, Iss2, Iss5, Iss7, Iss12, Iss22, Iss23, Iss26, Iss29, Iss34, Iss37, Iss43, Iss45, Iss50, Iss51, Iss53, Iss66, Iss68, Iss69, Iss72, Iss75, Iss84, Iss89, Iss95, Iss96, Iss99, Iss105, Iss107, Iss110, Iss113, Iss115, Iss117, Iss119, Iss120, Iss124, Iss126, Iss128, Iss129, Iss132, Iss133, Iss142, Iss146 and Iss150. The 43 issues are from the already identified 7 categories. The issues having equal means have been shown in the form of shaded blocks in Table 4.

An analogous method for determining the cut-off value is calculating the average of all means.

**4.4.2. Cut-off value based on average of all means.** Table 4 presents 'means of response values' for all the 150 issues.

Average of the means for all the 150 issues = 2.286084

By contemplating this average as cut-off value, once again the same first 43 issues from Table 4 qualify as the customarily arising issues.

Table 5 presents the 43 customarily issues together with the IDs, relevant means and respective categories.

From Table 5, we can get the answer to RQ2.

Table 5 shows that out of the 43 customarily arising issues, six issues belong to 'communication' category and seven issues belong to 'knowledge management & awareness' category. Similarly, 'cultural diversities' category causes five issues. Furthermore, five issues belong to 'management & coordination'. 'Processes & tools' category has five issues, six issues are related to 'relationship among stakeholders' whereas nine issues are 'requirements centric'.

Fig 4 pictorially shows no. of the customarily arising issues for every category.

**Table 4. Means, in descending order, of response values for 150 issues after 3<sup>rd</sup> round of Delphi technique.**

| Sr. # | Issue IDs | Means | Sr. # | Issue IDs | Means |
|---|---|---|---|---|---|
| 1. | Iss7 | 4.213592 | 32. | Iss150 | 3.990291 |
| 2. | Iss2 | 4.203883 | 33. | Iss132 | 3.990291 |
| 3. | Iss22 | 4.194175 | 34. | Iss113 | 3.990291 |
| 4. | Iss34 | 4.165049 | 35. | Iss110 | 3.970874 |
| 5. | Iss72 | 4.165049 | 36. | Iss66 | 3.970874 |
| 6. | Iss26 | 4.165049 | 37. | Iss126 | 3.970874 |
| 7. | Iss89 | 4.145631 | 38. | Iss117 | 3.961165 |
| 8. | Iss5 | 4.126214 | 39. | Iss96 | 3.922330 |
| 9. | Iss1 | 4.116505 | 40. | Iss115 | 3.922330 |
| 10. | Iss75 | 4.106796 | 41. | Iss107 | 3.854369 |
| 11. | Iss45 | 4.106796 | 42. | Iss23 | 3.854369 |
| 12. | Iss12 | 4.097087 | 43. | Iss119 | 3.825243 |
| 13. | Iss29 | 4.087379 | 44. | Iss148 | 2.019417 |
| 14. | Iss37 | 4.077670 | 45. | Iss42 | 2.000000 |
| 15. | Iss133 | 4.077670 | 46. | Iss41 | 1.961165 |
| 16. | Iss69 | 4.077670 | 47. | Iss8 | 1.912621 |
| 17. | Iss146 | 4.077670 | 48. | Iss108 | 1.873786 |
| 18. | Iss43 | 4.077670 | 49. | Iss61 | 1.815534 |
| 19. | Iss84 | 4.058252 | 50. | Iss20 | 1.796117 |
| 20. | Iss124 | 4.038835 | 51. | Iss31 | 1.766990 |
| 21. | Iss142 | 4.029126 | 52. | Iss28 | 1.766990 |
| 22. | Iss129 | 4.029126 | 53. | Iss149 | 1.728155 |
| 23. | Iss105 | 4.029126 | 54. | Iss16 | 1.728155 |
| 24. | Iss128 | 4.019417 | 55. | Iss143 | 1.679612 |
| 25. | Iss68 | 4.019417 | 56. | Iss64 | 1.669903 |
| 26. | Iss99 | 4.009709 | 57. | Iss4 | 1.669903 |
| 27. | Iss53 | 4.009709 | 58. | Iss67 | 1.660194 |
| 28. | Iss50 | 4.009709 | 59. | Iss9 | 1.660194 |
| 29. | Iss120 | 4.009709 | 60. | Iss81 | 1.660194 |
| 30. | Iss95 | 4.000000 | 61. | Iss17 | 1.660194 |
| 31. | Iss51 | 4.000000 | 62. | Iss13 | 1.660194 |
| 63. | Iss104 | 1.650485 | 97. | Iss35 | 1.533981 |
| 64. | Iss91 | 1.650485 | 98. | Iss11 | 1.533981 |
| 65. | Iss85 | 1.650485 | 99. | Iss112 | 1.533981 |
| 66. | Iss123 | 1.640777 | 100. | Iss94 | 1.533981 |
| 67. | Iss97 | 1.640777 | 101. | Iss87 | 1.533981 |
| 68. | Iss15 | 1.640777 | 102. | Iss79 | 1.533981 |
| 69. | Iss139 | 1.640777 | 103. | Iss76 | 1.533981 |
| 70. | Iss52 | 1.640777 | 104. | Iss134 | 1.533981 |
| 71. | Iss32 | 1.640777 | 105. | Iss131 | 1.533981 |
| 72. | Iss39 | 1.631068 | 106. | Iss102 | 1.533981 |
| 73. | Iss36 | 1.631068 | 107. | Iss77 | 1.524272 |
| 74. | Iss27 | 1.631068 | 108. | Iss135 | 1.524272 |
| 75. | Iss55 | 1.621359 | 109. | Iss127 | 1.524272 |
| 76. | Iss46 | 1.601942 | 110. | Iss92 | 1.524272 |
| 77. | Iss116 | 1.582524 | 111. | Iss63 | 1.524272 |
| 78. | Iss147 | 1.572816 | 112. | Iss70 | 1.524272 |

*(Continued)*

**Table 4.** (Continued)

| Sr. # | Issue IDs | Means | Sr. # | Issue IDs | Means |
|---|---|---|---|---|---|
| 79. | Iss138 | 1.563107 | 113. | Iss90 | 1.514563 |
| 80. | Iss144 | 1.563107 | 114. | Iss60 | 1.514563 |
| 81. | Iss109 | 1.563107 | 115. | Iss57 | 1.514563 |
| 82. | Iss103 | 1.563107 | 116. | Iss54 | 1.514563 |
| 83. | Iss3 | 1.563107 | 117. | Iss40 | 1.514563 |
| 84. | Iss83 | 1.553398 | 118. | Iss49 | 1.514563 |
| 85. | Iss24 | 1.553398 | 119. | Iss93 | 1.504854 |
| 86. | Iss101 | 1.553398 | 120. | Iss65 | 1.504854 |
| 87. | Iss73 | 1.553398 | 121. | Iss56 | 1.504854 |
| 88. | Iss140 | 1.553398 | 122. | Iss38 | 1.504854 |
| 89. | Iss121 | 1.543689 | 123. | Iss18 | 1.504854 |
| 90. | Iss130 | 1.543689 | 124. | Iss145 | 1.504854 |
| 91. | Iss106 | 1.543689 | 125. | Iss33 | 1.504854 |
| 92. | Iss58 | 1.543689 | 126. | Iss30 | 1.504854 |
| 93. | Iss47 | 1.543689 | 127. | Iss25 | 1.504854 |
| 94. | Iss19 | 1.543689 | 128. | Iss137 | 1.504854 |
| 95. | Iss14 | 1.543689 | 129. | Iss118 | 1.504854 |
| 96. | Iss74 | 1.543689 | 130. | Iss114 | 1.504854 |
| 131. | Iss111 | 1.504854 | 141. | Iss59 | 1.495146 |
| 132. | Iss100 | 1.504854 | 142. | Iss141 | 1.485437 |
| 133. | Iss136 | 1.495146 | 143. | Iss6 | 1.485437 |
| 134. | Iss125 | 1.495146 | 144. | Iss98 | 1.485437 |
| 135. | Iss21 | 1.495146 | 145. | Iss88 | 1.485437 |
| 136. | Iss10 | 1.495146 | 146. | Iss82 | 1.485437 |
| 137. | Iss71 | 1.495146 | 147. | Iss78 | 1.485437 |
| 138. | Iss86 | 1.495146 | 148. | Iss48 | 1.485437 |
| 139. | Iss80 | 1.495146 | 149. | Iss44 | 1.485437 |
| 140. | Iss62 | 1.495146 | 150. | Iss122 | 1.475728 |

## 4.5. Ranking the ccustomarily arising issues category-wise

The study ranks the customarily arising SDO RE process issues category-wise depending on the means of issues like preceding studies [81–83]. The criterion employed for the ranking is 'frequency of occurrence' of the issues.

**4.5.1. Ranks of the ccustomarily arising communication issues.** By capitalizing on the details given in Table 5, Table 6 introduces the means of the customarily arising communication issues in the descending order. Hinging on the issues' means, the category-wise ranks of the issues can be determined. Table 6 also presents, the average for the means of all the six customarily arising communication issues which is 4.158576.

**4.5.2. Ranks of the ccustomarily arising knowledge management and awareness issues.** By capitalizing on the details given in Table 5, Table 7 introduces the means of the customarily arising knowledge management and awareness issues in the descending order. Hinging on the issues' means, the category-wise ranks of the issues can be determined. Table 7 also presents, the average for the means of all the seven customarily arising knowledge management and awareness issues which is 4.076283.

**4.5.3. Ranks of the customarily arising cultural diversities issues.** By capitalizing on the details given in Table 5, Table 8 introduces the means of the customarily arising cultural

**Table 5. Frequently or customarily arising issues of RE process for SDO along with respective means and categories.**

| Sr. # | Frequently or customarily arising SDO RE issues and IDs | Means | Categories |
|---|---|---|---|
| 1 | Iss1: Occasional and controlled correspondence amongst the shareholders [40]. | 4.116505 | Communication |
| 2 | Iss2: Deficiency of casual correspondence amongst the shareholders [33, 91–93]. | 4.203883 | |
| 3 | Iss5: Deficiency of synchronized correspondence [96–97]. | 4.126214 | |
| 4 | Iss7: Deferred replies [93, 99–100]. | 4.213592 | |
| 5 | Iss12: The gatherings that are held for making decisions regarding requirements are fruitless [28,33]. | 4.097087 | |
| 6 | Iss22: Typically, there is non-recording of the promises that are done amid videoconferencing or discussions on the telephone, consequently such pledges cannot be alluded when needed [Proposed]. | 4.194175 | |
| 7 | Iss23: Obstacles in flow of requirements information towards organizations or from organization [108]. | 3.854369 | Knowledge management and awareness |
| 8 | Iss26: Unfamiliarity of the shareholders from existing/recent data regarding requirements [111]. | 4.165049 | |
| 9 | Iss29: Reviving of the previously conversed and apparently resolved issues [38, 113]. | 4.087379 | |
| 10 | Iss34: Inadequate management of the modifications in requirements [69, 115]. | 4.165049 | |
| 11 | Iss37: Functioning on the outdated requirements [111, 117]. | 4.077670 | |
| 12 | Iss43: Requirements engineers are ignorant of the impacts of novel system deployment upon customer's organization [121]. | 4.077670 | |
| 13 | Iss45: Unfamiliarity with or not consulting all the origins of requirements [Proposed]. | 4.106796 | |
| 14 | Iss50: Scarcity of trust amongst the different shareholders [17, 93, 107, 122–123, 126]. | 4.009709 | Cultural diversities |
| 15 | Iss51: Evasion of the obligations from the different shareholders [94]. | 4.000000 | |
| 16 | Iss53: Complications in attaining consent on requirements [30, 40, 94]. | 4.009709 | |
| 17 | Iss66: Noninvolvement or elimination of shareholders during RE related events [Proposed]. | 3.970874 | |
| 18 | Iss68: Challenges to set the practical assumptions regarding reply time [Proposed]. | 4.019417 | |
| 19 | Iss69: Complications in grasping evidences, motives and actions needed for mutual Requirements Understanding (RU) amongst the scattered shareholders [29, 33, 102]. | 4.077670 | Management and coordination |
| 20 | Iss72: Postponement in elucidations regarding requirements and finalizing decisions [94]. | 4.165049 | |
| 21 | Iss75: Improperly defined or vague obligations [118, 135]. | 4.106796 | |
| 22 | Iss84: Genuine requirements are needed to be altered to interface with different software systems [135]. | 4.058252 | |
| 23 | Iss89: Failure in performing RE associated assignment(s) as everyone believes this is obligation of another person [Proposed]. | 4.145631 | |
| 24 | Iss95: Utilization of various RE procedures introduces various formats and techniques at distant sites of customer [26, 136]. | 4.000000 | Process and tools |
| 25 | Iss96: Utilizing inappropriate RE procedures [118]. | 3.922330 | |
| 26 | Iss99: RE associated rework or information loss amid exchanges among various tools [26]. | 4.009709 | |
| 27 | Iss105: Choosing the unsuitable RE instrument(s) [26, 118]. | 4.029126 | |
| 28 | Iss107: Utilization of inadequate technique for eliciting requirements [Proposed]. | 3.854369 | |
| 29 | Iss110: Absence of steady relationship amongst the shareholders [93, 141]. | 3.970874 | Relationship among stakeholders |
| 30 | Iss113: Utilization of various standards, by client and vendor, for documenting the requirements [26]. | 3.990291 | |
| 31 | Iss115: Disparate preferences of customer and vendor to collect and confirm requirements [26]. | 3.922330 | |
| 32 | Iss117: Team(s) from vendor side have misapprehensions regarding working practices of the client side [26]. | 3.961165 | |
| 34 | Iss120: Problems of deciding about requirements related deliverables [26]. | 4.009709 | |
| 35 | Iss124: Confirming requirements in case of all shareholders relying on the requirements collected or data acquired only from the accessible shareholders [129]. | 4.038835 | Requirements centric |
| 36 | Iss126: Inaccurate or wrong requirements [143]. | 3.970874 | |
| 37 | Iss128: Gold-plated or additional requirements [144]. | 4.019417 | |
| 38 | Iss129: Uncompleted requirements [109, 137, 143]. | 4.029126 | |
| 39 | Iss132: Requirements are described/specified ambiguously [5, 21, 69, 109, 118, 146]. | 3.990291 | |
| 40 | Iss133: Not giving data or giving deliberately vague data about requirements [33, 102]. | 4.077670 | |
| 41 | Iss142: Analysts are influenced to conceal certain data associated to requirements that grounds for compromises to elicit and describe the requirements [121]. | 4.029126 | |
| 42 | Iss146: Customers emphasis on including more requirements whereas cost and schedule have been settled [Proposed]. | 4.077670 | |
| 43 | Iss150: Applying presumptions to confirm or conclude requirements [Proposed]. | 3.990291 | |

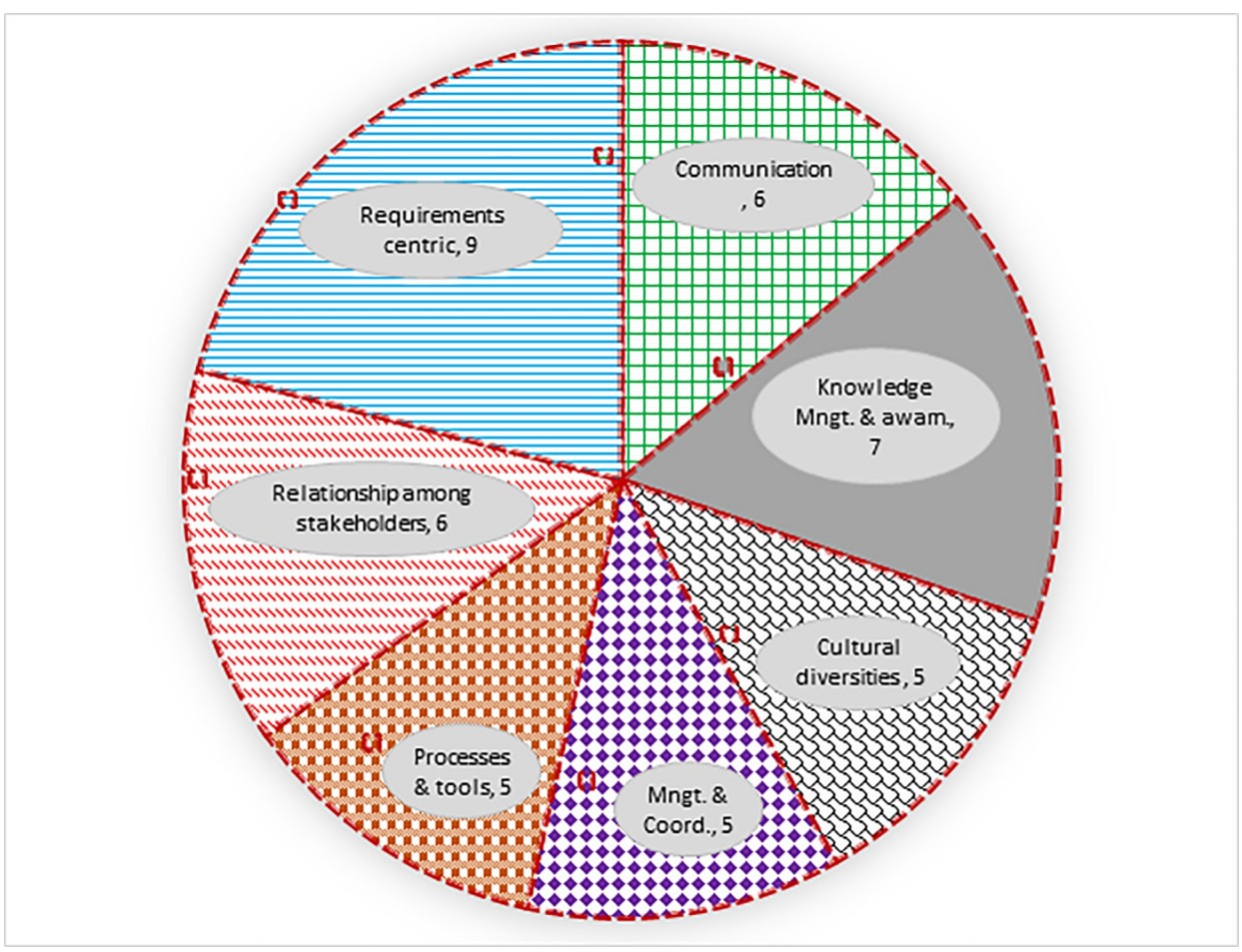

**Fig 4. No. of ccustomarily arising issues in case of each category.**

diversities' issues in the descending order. Hinging on the issues' means, the category-wise ranks of the issues can be determined. Table 8 also presents, the average for the means of all the five customarily arising cultural diversities issues which is 4.001942.

**4.5.4. Ranks of the ccustomarily arising management and coordination issues.** By capitalizing on the details given in Table 5, Table 9 introduces the means of the customarily arising management and coordination issues in the descending order. Hinging on the issues' means, the category-wise ranks of the issues can be determined. Table 9 also presents, the average for

**Table 6. Ranks of communication issues.**

| Sr. # | Issue IDs | Means in descending order | Category wise ranks |
|---|---|---|---|
| 1 | Iss7 | 4.213592 | 1 |
| 2 | Iss2 | 4.203883 | 2 |
| 3 | Iss22 | 4.194175 | 3 |
| 4 | Iss5 | 4.126214 | 4 |
| 5 | Iss1 | 4.116505 | 5 |
| 6 | Iss12 | 4.097087 | 6 |
| Average of the means of communication issues | | | 4.158576 |

Table 7. Ranks of knowledge management and awareness issues.

| Sr. # | Issue IDs | Means in descending order | Category wise ranks |
|---|---|---|---|
| 1 | Iss34 | 4.165049 | 1 |
| 2 | Iss26 | 4.165049 | 2 |
| 3 | Iss45 | 4.106796 | 3 |
| 4 | Iss29 | 4.087379 | 4 |
| 5 | Iss43 | 4.077670 | 5 |
| 6 | Iss37 | 4.077670 | 6 |
| 7 | Iss23 | 3.854369 | 7 |
| Average of the means of knowledge management and awareness issues | | | 4.076283 |

the means of all the five customarily arising management and coordination issues which is 4.110680.

**4.5.5. Ranks of the customarily arising processes and tools issues.** By capitalizing on the details given in Table 5, Table 10 introduces the means of the customarily arising processes and tools' issues in the descending order. Hinging on the issues' means, the category-wise ranks of the issues can be determined. Table 10 also presents, the average for the means of all the five customarily arising processes and tools' issues which is 3.963107.

**4.5.6. Ranks of the customarily arising relationship among stakeholders' issues.** By capitalizing on the details given in Table 5, Table 11 introduces the means of the customarily arising relationship among stakeholders' issues in the descending order. Hinging on the issues' means, the category-wise ranks of the issues can be determined. Table 11 also presents, the average for the means of all the six customarily arising relationship among stakeholders' issues which is 3.946602.

**4.5.7. Ranks of the customarily arising requirements centric issues.** By capitalizing on the details given in Table 5, Table 12 introduces the means of the customarily arising requirements centric issues in the descending order. Hinging on the issues' means, the category-wise ranks of the issues can be determined. Table 12 also presents, the average for the means of all the nine customarily arising requirements centric issues which is 4.024811.

Tables 6–12 present the ranks of customarily arising issues within their corresponding categories and hence provide the answer to RQ3.1.

## 4.6. Overall ranks of the customarily arising issues

By capitalizing on the details given in Table 5, Table 13 introduces the means of the 43 customarily arising issues in the descending order. Hinging on the issues' means, the overall ranks of the 43 customarily arising issues can be determined.

This provides the answer to RQ3.2.

Table 8. Ranks of cultural diversities issues.

| Sr. # | Issue IDs | Means in descending order | Category wise ranks |
|---|---|---|---|
| 1 | Iss68 | 4.019417 | 1 |
| 2 | Iss53 | 4.009709 | 2 |
| 3 | Iss50 | 4.009709 | 3 |
| 4 | Iss51 | 4.000000 | 4 |
| 5 | Iss66 | 3.970874 | 5 |
| Average of the means of cultural diversities' issues | | | 4.001942 |

Table 9. Ranks of management and coordination issues.

| Sr. # | Issue IDs | Means in descending order | Category wise ranks |
|---|---|---|---|
| 1 | Iss72 | 4.165049 | 1 |
| 2 | Iss89 | 4.145631 | 2 |
| 3 | Iss75 | 4.106796 | 3 |
| 4 | Iss69 | 4.077670 | 4 |
| 5 | Iss84 | 4.058252 | 5 |
| Average of the means of management and coordination issues | | | 4.110680 |

## 4.7. Ranking the categories of the customarily arising issues

By capitalizing on the details given in the ending rows of Tables 6–12; Table 14 introduces the means for the various categories of the issues.

By capitalizing on the details given in Table 14, Table 15 introduces the means for the various categories of the issues in the descending order. Hinging on the categories' means, the ranks of the issues' categories can be determined.

This provides the answer to RQ3.3

## 4.8. Putting category-wise ranks, overall ranks and categories' ranks together

Table 16 presents the 43 customarily arising SDO RE process issues in conjunction with individual ranks of the issues with respect to the corresponding categories. The overall ranks of the 43 customarily arising issues as well as ranks of the seven categories of the customarily arising issues, have also been delineated. The 43 customarily arising issues have been articulated by the notations $I_1$, $I_2$, $I_3$, . . ., $I_{43}$ respectively.

This provides the answer to RQ3 as a whole.

## 5. Discussion

Firstly, this study identifies the categories for the issues of RE process in the case of SDO. The nine potential categories are: i. Communication, ii. Knowledge management and awareness, iii. Cultural diversities, iv. Trust, v. Management and coordination, vi. Organizational structure, vii. Processes and tools, viii. Relationship among stakeholders, and ix. Requirements centric. Based on a questionnaire survey with the SDO industry practitioners and by applying 50% rule, the seven categories, except trust and organizational structure, have been selected as the categories for the issues of RE process in the case of SDO. At least 50% or more survey participants have selected these seven categories as the categories for the issues of RE process in the case of SDO.

Table 10. Ranks of processes and tools issues.

| Sr. # | Issue IDs | Means in descending order | Category wise ranks |
|---|---|---|---|
| 1 | Iss105 | 4.029126 | 1 |
| 2 | Iss99 | 4.009709 | 2 |
| 3 | Iss95 | 4.000000 | 3 |
| 4 | Iss96 | 3.922330 | 4 |
| 5 | Iss107 | 3.854369 | 5 |
| Average of the means of processes and tools' issues | | | 3.963107 |

**Table 11. Ranks of relationship among stakeholders' issues.**

| Sr. # | Issue IDs | Means in descending order | Category wise ranks |
|---|---|---|---|
| 1 | Iss120 | 4.009709 | 1 |
| 2 | Iss113 | 3.990291 | 2 |
| 3 | Iss110 | 3.970874 | 3 |
| 4 | Iss117 | 3.961165 | 4 |
| 5 | Iss115 | 3.922330 | 5 |
| 6 | Iss119 | 3.825243 | 6 |
| Average of the means of relationship among stakeholders' issues | | | 3.946602 |

The study also explores 150 issues for the SDO RE process. The 129 issues have been extracted from the contemporary literature whereas 21 issues have been identified from the SDO industry. For the literature assessment, 2335 studies have been retrieved from the five selected electronic databases: i. IEEE Xplore, ii. ACM, iii. Science Direct, iv. Springer Link and v. Web of Science. Out of 2335 studies, 77 studies have been selected finally for the further analysis. To explore the SDO RE process issues that are faced by the SDO industry practitioners, a questionnaire survey has been conducted by involving SDO practitioners and 21 issues have been identified. Out of the 150 (129+21) issues, there are 22 communication issues, 24 knowledge management and awareness issues, 22 cultural diversities issues, 22 management and coordination issues, 19 processes and tools issues, 14 relationship among stakeholders' issues and 27 requirements centric issues. The succeeding subsection presents category-wise complete list of 150 issues.

**i. Communication issues**, Iss1: Occasional and controlled correspondence amongst the shareholders [40], Iss2: Deficiency of casual correspondence amongst the shareholders [33, 91–93], Iss3: To explain and resolve the confusions regarding requirements, person to person correspondence is essential [94], Iss4: Deficiency of person to person correspondence [93, 95], Iss5:Deficiency of synchronized correspondence [96–97, Iss6: Even via the videoconferences, it is difficult to enable extensive and fruitful dialogs specifically in case of numerous shareholders [98], Iss7: Deferred replies [93, 99–100],

Iss8: Planning the co-located gatherings amongst shareholders is impractical mostly as shareholders are detached [101–102], Iss9: There is improper correspondence between customer and vendor [69],

Iss10:Organizing person to person get-togethers heightens the cost [21, 97, 101], Iss11: Shareholders do not utilize synchronized Internet communication technologies to convey

**Table 12. Ranks of requirements centric issues.**

| Sr. # | Issue IDs | Means in descending order | Category wise ranks |
|---|---|---|---|
| 1. | Iss146 | 4.077670 | 1 |
| 2. | Iss133 | 4.077670 | 2 |
| 3. | Iss124 | 4.038835 | 3 |
| 4. | Iss142 | 4.029126 | 4 |
| 5. | Iss129 | 4.029126 | 5 |
| 6. | Iss128 | 4.019417 | 6 |
| 7. | Iss150 | 3.990291 | 7 |
| 8. | Iss132 | 3.990291 | 8 |
| 9. | Iss126 | 3.970874 | 9 |
| Average of the means of requirements centric issues | | | 4.024811 |

**Table 13. Overall ranks of the 43 customarily arising issues of SDO RE process.**

| Sr. # | Issue IDs | Means | Overall ranks |
|---|---|---|---|
| 1. | Iss7 | 4.213592 | 1 |
| 2. | Iss2 | 4.203883 | 2 |
| 3. | Iss22 | 4.194175 | 3 |
| 4. | Iss34 | 4.165049 | 4 |
| 5. | Iss72 | 4.165049 | 4 |
| 6. | Iss26 | 4.165049 | 4 |
| 7. | Iss89 | 4.145631 | 7 |
| 8. | Iss5 | 4.126214 | 8 |
| 9. | Iss1 | 4.116505 | 9 |
| 10. | Iss75 | 4.106796 | 10 |
| 11. | Iss45 | 4.106796 | 10 |
| 12. | Iss12 | 4.097087 | 12 |
| 13. | Iss29 | 4.087379 | 13 |
| 14. | Iss37 | 4.077670 | 14 |
| 15. | Iss133 | 4.077670 | 14 |
| 16. | Iss69 | 4.077670 | 14 |
| 17. | Iss146 | 4.077670 | 14 |
| 18. | Iss43 | 4.077670 | 14 |
| 19. | Iss84 | 4.058252 | 19 |
| 20. | Iss124 | 4.038835 | 20 |
| 21. | Iss142 | 4.029126 | 21 |
| 22. | Iss129 | 4.029126 | 21 |
| 23. | Iss105 | 4.029126 | 21 |
| 24. | Iss128 | 4.019417 | 24 |
| 25. | Iss68 | 4.019417 | 24 |
| 26. | Iss99 | 4.009709 | 26 |
| 27. | Iss53 | 4.009709 | 26 |
| 28. | Iss50 | 4.009709 | 26 |
| 29. | Iss120 | 4.009709 | 26 |
| 30. | Iss95 | 4.000000 | 30 |
| 31. | Iss51 | 4.000000 | 30 |
| 32. | Iss150 | 3.990291 | 32 |
| 33. | Iss132 | 3.990291 | 32 |
| 34. | Iss113 | 3.990291 | 32 |
| 35. | Iss110 | 3.970874 | 35 |
| 36. | Iss66 | 3.970874 | 35 |
| 37. | Iss126 | 3.970874 | 35 |
| 38. | Iss117 | 3.961165 | 38 |
| 39. | Iss96 | 3.922330 | 39 |
| 40. | Iss115 | 3.922330 | 39 |
| 41. | Iss107 | 3.854369 | 41 |
| 42. | Iss23 | 3.854369 | 41 |
| 43. | Iss119 | 3.825243 | 43 |

information regarding requirements, instead rely on traditional approaches alike planned meetings, electronic mails and documentation [92], Iss12: The gatherings that are held for

Table 14. Means in case of the 7 categories of SDO RE process issues.

| Sr. # | Categories | Means |
|-------|------------|-------|
| 1 | Communication | 4.158576 |
| 2 | Knowledge management & awareness | 4.076283 |
| 3 | Cultural diversities | 4.001942 |
| 4 | Management and coordination | 4.110680 |
| 5 | Processes and tools | 3.963107 |
| 6 | Relationship among stakeholders | 3.946602 |
| 7 | Requirements centric | 4.024811 |

making decisions regarding requirements are fruitless [28,33], Iss13: Asynchronous correspondence leads to deferment in proliferation and resolution of issues [102], Iss14: If there are synchronous meetings amid the locations which have substantial differences regarding time then participants belonging to some locations are bothered as there are huge differences between the meeting times and their local working times [40, 102–103], Iss15: Shareholders are not able to express in the correspondence language [33], Iss16: Electronic correspondence alike email permits clandestine correspondence that generates complications for settling clashes regarding requirements [33], Iss17: Shareholders don't convey to one another adequately, instead seek to apply force and utilize influence on one another [102], Iss18: To illuminate and resolve the issues, any coworker may correspond with any shareholder that may cause tedious debates and additional controlling endeavors [104], Iss19: Correspondence gaps or postponements amid RE because of individuality conflicts [105], Iss20: Online correspondence to elucidate requirements prompts spiny requirements because resulting requirements are uncertain, alter again and again or are unfinished [106], Iss21: To arrange interviews, acquiring the assent of far off shareholders [107], Iss22: Typically, there is non-recording of the promises that are done amid videoconferencing or discussions on the telephone, consequently such pledges cannot be alluded when needed [Proposed].

**ii. Knowledge management & awareness issues**, Iss23: Obstacles in flow of requirements information towards organizations or from organization [108], Iss24: Ineptitude of keeping track of the shareholders, and related data, who are influenced because of the introduction of novel requirements [109], Iss25: Shareholders are incompetent to look for pertinent information, strategies are coordinated improperly to incorporate the information, and information exchange is deferred or blocked[110], Iss26: Unfamiliarity of the shareholders from existing/recent data regarding requirements [111], Iss27: Requirements data attained by various far off origins is not imparted to every shareholder [28, 40, 103], Iss28: Physically dispersed shareholders are unable to receive the rewards of communal mechanisms and procedures that are available for collocated workspace, consequently, need for consciousness regarding the requirements is increased [112], Iss29: Reviving of the previously conversed and apparently resolved issues [38, 113], Iss30: Inappropriate allocation of duties, with respect to administrative organization, may hamper the circulation of information [114], Iss31: Proliferation of the data regarding requirements modifications is inadequate [92], Iss32: Professionals inadvertently neglect to apprise pertinent shareholders regarding the modifications in requirements [92], Iss33: The professionals' clusters engaged in the similar or related requirements are unaware about the shareholders who are influenced by requirements modifications or who stimulate the requirements modifications [92], Iss34: Inadequate management of the modifications in requirements [69, 115], Iss35: The diversified bunches engaged in similar or linked requirements are uninformed regarding the specialists of the far-off teams [92], Iss36: Traditional sources for correspondence alike documents are unable to reveal the alterations in

**Table 15. Ranks of the categories of customarily arising SDO RE process issues.**

| Sr. # | Categories | Means in descending order | Categories' ranks |
|---|---|---|---|
| 1 | Communication | 4.158576 | 1 |
| 2 | Management and coordination | 4.110680 | 2 |
| 3 | Knowledge management and awareness | 4.076283 | 3 |
| 4 | Requirements centric | 4.024811 | 4 |
| 5 | Cultural diversities | 4.001942 | 5 |
| 6 | Processes and tools | 3.963107 | 6 |
| 7 | Relationship among stakeholders | 3.946602 | 7 |

requirements as fast as needed [112, 116], Iss37: Functioning on the outdated requirements [111, 117], Iss38: Hitches in accessibility of the steady data because of the dissemination of sources [118], Iss39: Scarcity of the mindfulness regarding deployment environment may cause ambiguity in requirements [94], Iss40: Unfamiliarity to the background and significance of requirements can cause project postponements and quality tradeoffs [119], Iss41: Requirements illuminations are passed on later than expected time which can cause project postponements [111], Iss42: Incapability to share information or finest practices [28, 120], Iss43: Requirements engineers are ignorant of the impacts of novel system deployment upon customer's organization [121], Iss44: The professional bunches engaged in similar or related requirements do not know which requirement is being addressed by whom [Proposed], Iss45: Unfamiliarity with or not consulting all the origins of requirements [Proposed], Iss46: Inappropriate tracking of the requirements [Proposed].

**iii. Cultural diversities' issues**, Iss47: Detachment leads to cultural variances amongst the different working departments belonging to an organization which produces difficulty in achieving the shared awareness about the requirements[33, 102], Iss48: Generating trust amongst the different shareholders is demanding [33, 107, 122–125], Iss49: Upholding trust amongst the different shareholders is demanding [123, 125], Iss50: Scarcity of trust amongst the different shareholders [17, 93, 107, 122–123, 126], Iss51: Evasion of the obligations from the different shareholders [94], Iss52: Forfeiture of attachment amongst the shareholders on account of physical dispersal [127], Iss53: Complications in attaining consent on requirements [30, 40, 94, 128], Iss54: Shareholders originate from miscellaneous social backgrounds and own dissimilar moral standards regarding hierarchies, addressing risks, tracking timetables and promptness that can intensify disagreements [94], Iss55: Various cultures follow dissimilar values concerning exactness of work done and capability of inventiveness [118], Iss56: Professionals from differing social foundations have ambiguous and implicit implications and clarifications of the data about the requirements[39, 129], Iss57: Professionals from different social foundations derive mixt implications from messages [130], Iss58: A few experts, due to their social foundations, cannot do disagreement with the customers, hence, 'pleasing' requirements and main requirements are assigned same preferences [118], Iss59: Requirements of the client are not completely comprehended and conveyed due to divergent cultural foundations and languages [131], Iss60: Contributors of the far-off gatherings, regarding requirements engineering, are not skilled in sole communication language [97, 132], Iss61: Shareholders are at various capability level of the correspondence language, consequently, shareholders at advanced level influence and dominate the correspondence about requirements [100], Iss62: Identical words are utilized to pass on the dissimilar implications in various associations that generates confusions for requirements description and approval [33], Iss63: The persons, not capable in correspondence language, are hesitant in making inquiries for requirements elucidations [100], Iss64: Bashfulness of the shareholders, for instance evading from doing

**Table 16. Ranks of the customarily arising issues of SDO RE process and ranks of the issues' categories.**

| Sr. # | SDO RE process issues and IDs | Issues' ranks | | Categories' | |
|---|---|---|---|---|---|
| | | Category-wise | Overall | Ranks | Names |
| $I_1$ | Iss7: Deferred replies [93, 99–100]. | 1 | 1 | 1 | Communication |
| $I_2$ | Iss2: Deficiency of casual correspondence amongst the shareholders [33, 91–93]. | 2 | 2 | | |
| $I_3$ | Iss22: Typically, there is non-recording of the promises that are done amid videoconferencing or discussions on the telephone, consequently such pledges cannot be alluded when needed [Proposed]. | 3 | 3 | | |
| $I_4$ | Iss5: Deficiency of synchronized correspondence [96–97]. | 4 | 8 | | |
| $I_5$ | Iss1: Occasional and controlled correspondence amongst the shareholders [40]. | 5 | 9 | | |
| $I_6$ | Iss12: The gatherings that are held for making decisions regarding requirements are fruitless [28, 33]. | 6 | 12 | | |
| $I_7$ | Iss72: Postponement in elucidations regarding requirements and finalizing decisions [94]. | 1 | 4 | 2 | Management and coordination |
| $I_8$ | Iss89: Failure in performing RE associated assignment(s) as everyone believes this is obligation of another person [Proposed]. | 2 | 7 | | |
| $I_9$ | Iss75: Improperly defined or vague obligations [118, 135]. | 3 | 10 | | |
| $I_{10}$ | Iss69: Complications in grasping evidences, motives and actions needed for mutual Requirements Understanding (RU) amongst the scattered shareholders [29, 33, and 102]. | 4 | 14 | | |
| $I_{11}$ | Iss84: Genuine requirements are needed to be altered to interface with different software systems [135]. | 5 | 19 | | |
| $I_{12}$ | Iss34: Inadequate management of the modifications in requirements [69, 115]. | 1 | 4 | 3 | Knowledge management and awareness |
| $I_{13}$ | Iss26: Unfamiliarity of the shareholders from existing/recent data regarding requirements [111]. | 2 | 4 | | |
| $I_{14}$ | Iss45: Unfamiliarity with or not consulting all the origins of requirements [Proposed]. | 3 | 10 | | |
| $I_{15}$ | Iss29: Reviving of the previously conversed and apparently resolved issues [38, 113]. | 4 | 13 | | |
| $I_{16}$ | Iss43: Requirements engineers are ignorant of the impacts of novel system deployment upon customer's organization [121]. | 5 | 14 | | |
| $I_{17}$ | Iss37: Functioning on the outdated requirements [111, 117]. | 6 | 14 | | |
| $I_{18}$ | Iss23: Obstacles in flow of requirements related information towards organizations or from organization [108]. | 7 | 41 | | |
| $I_{19}$ | Iss146: Customers emphasis on including more requirements whereas cost and schedule have been settled [Proposed]. | 1 | 14 | 4 | Requirements centric |
| $I_{20}$ | Iss133: Not giving data or giving deliberately vague data about requirements [33, 102]. | 2 | 14 | | |
| $I_{21}$ | Iss124: Confirming requirements in case of all shareholders relying on the requirements collected or data acquired only from the accessible shareholders [129]. | 3 | 20 | | |
| $I_{22}$ | Iss142: Analysts are influenced to conceal certain data associated to requirements that grounds for compromises to elicit and describe the requirements [121]. | 4 | 21 | | |
| $I_{23}$ | Iss129: Uncompleted requirements [109, 137, 143]. | 5 | 21 | | |
| $I_{24}$ | Iss128: Gold-plated or additional requirements [144]. | 6 | 24 | | |
| $I_{25}$ | Iss150: Applying presumptions to confirm or conclude requirements [Proposed]. | 7 | 32 | | |
| $I_{26}$ | Iss132: Requirements are described/specified ambiguously [5, 21, 69, 109, 118, 146]. | 8 | 32 | | |
| $I_{27}$ | Iss126: Inaccurate or wrong requirements [143]. | 9 | 35 | | |
| $I_{28}$ | Iss68: Challenges to set the practical assumptions regarding reply time [Proposed]. | 1 | 24 | 5 | Cultural diversities |
| $I_{29}$ | Iss53: Complications in attaining consent on requirements [30, 40, 94, 128]. | 2 | 26 | | |
| $I_{30}$ | Iss50: Scarcity of trust amongst the different shareholders [17, 93, 107, 122–123, 126]. | 3 | 26 | | |
| $I_{31}$ | Iss51: Evasion of the obligations from the different shareholders [94]. | 4 | 30 | | |
| $I_{32}$ | Iss66: Noninvolvement or elimination of shareholders during RE related events [Proposed]. | 5 | 35 | | |

(*Continued*)

**Table 16.** (Continued)

| Sr. # | SDO RE process issues and IDs | Issues' ranks | | Categories' | |
|---|---|---|---|---|---|
| | | Category-wise | Overall | Ranks | Names |
| I₃₃ | Iss105: Choosing the unsuitable RE instrument(s) [26, 118]. | 1 | 21 | 6 | Processes and tools |
| I₃₄ | Iss99: RE associated rework or information loss amid exchanges among various tools [26]. | 2 | 26 | | |
| I₃₅ | Iss95: Utilization of various RE procedures introduces various formats and techniques at distant sites of customer [26, 136]. | 3 | 30 | | |
| I₃₆ | Iss96: Utilizing inappropriate RE procedures [118]. | 4 | 39 | | |
| I₃₇ | Iss107: Utilization of inadequate technique for eliciting requirements [Proposed]. | 5 | 41 | | |
| I₃₈ | Iss120: Problems of deciding about requirements related deliverables [26]. | 1 | 26 | 7 | Relationship among stakeholders |
| I₃₉ | Iss113: Utilization of various standards, by client and vendor, for documenting the requirements [26]. | 2 | 32 | | |
| I₄₀ | Iss110: Absence of steady relationship amongst the shareholders [93, 141]. | 3 | 35 | | |
| I₄₁ | Iss117: Team(s) from vendor side have misapprehensions regarding working practices of the client side [26]. | 4 | 38 | | |
| I₄₂ | Iss115: Disparate preferences of customer and vendor to collect and confirm requirements [26]. | 5 | 39 | | |
| I₄₃ | Iss119: Unsuccessfulness of vendor to meet due dates and satisfy the obligations regarding requirements [26]. | 6 | 43 | | |

telephone calls to unacquainted individuals, causes deferred correspondence [101], Iss65: The requirements cognizance is diminished in case of describing the requirements in the non-indigenous language [94], Iss66: Noninvolvement or elimination of shareholders during RE related events [Proposed], Iss67: A portion of the stakeholders do not take part in the RE associated discussions in view of their non-familiarity with the correspondence language [Proposed], Iss68: Challenges to set the practical assumptions regarding reply time [Proposed].

**iv. Management and coordination issues**, Iss69: Complications in grasping evidences, motives and actions needed for mutual Requirements Understanding (RU) amongst the scattered shareholders [29, 33, 102], Iss70: Disparities in the regional-times of the stakeholders create hindrance in synchronizing RE associated events [133–134], Iss71: Obstruction for contribution of shareholders in RE related events due to time contrasts [40], Iss72: Postponement in elucidations regarding requirements and finalizing decisions [94], Iss73: Tendency of not-mentioning RE-related issues due to remoteness [103], Iss74: Even the skillful experts can end up anxious and dormant on account of being far off [105], Iss75: Improperly defined or vague obligations [118, 135], Iss76: Absenteeism of pivotal and reliable administration for RE process that origins improper coordination [105], Iss77: Absenteeism of a steady, talented and focal analyst role [105], Iss78: Underrating the time needed for performing requirements appraisal [105], Iss79: Discriminating distribution of working load to different groups [136], Iss80: No evaluation of the impact of shareholders' dissemination on various RE related tasks [136], Iss81: Contradictory benefits of various shareholders[30, 33], Iss82: Requirements obtained from the distributed shareholders belonging to different hierarchical units, are needed to be bundled [135], Iss83: Requirements are obtained from the huge number of shareholders [118], Iss84: Genuine requirements are needed to be altered to interface with different software systems [135], Iss85: Requirements are modified by analyst by overlooking the recommended procedure [105], Iss86: Given the time-based dispersal, harmonized coordination is needed to generate the trust [134], Iss87: Distant RE groups work with confined timetable to fulfill deadlines [5, 137], Iss88: Group fellow(s) expect that other group fellow(s) have to accomplish similar obligations [Proposed], Iss89: Failure in performing RE associated

assignment(s) as everyone believes this is obligation of another person [Proposed], Iss90: Impractical resource division to accomplish RE [Proposed].

**v. Processes and tools' issues**, Iss91: Absence of obviously delineated RE process [94, 136], Iss92: The shareholders utilize divergent procedures for examining and recording requirements [92], Iss93: Shareholders utilize diverse procedures to conduct alterations in requirements [92], Iss94: The standard RE procedures are not followed [105, 118], Iss95: Utilization of various RE procedures introduces various formats and techniques at distant sites of customer [26, 136], Iss96: Utilizing inappropriate RE procedures [118], Iss97: Some group fellows don't participate in RE consultations because they are unfamiliar with the apparatuses and techniques being utilized [138], Iss98: The instruments can't be merged with different instruments [118], Iss99: RE associated rework or information loss amid exchanges among various tools [26], Iss100: Necessity for the instruments that give perpetual access to data associated with requirements [117], Iss101: Instruments don't pass on data, about requirements change, to the pertinent shareholders at the suitable time [139], Iss102: Necessity for the instruments that enable the discernibility of requirements crosswise the fringes of instruments [117], Iss103: Necessity for the instruments that assist requirements dialogs amongst the distant shareholders [140], Iss104: Incapability of the tools for evolving the requirements documents by enabling coordination amongst the distant shareholders [139], Iss105: Choosing the unsuitable RE instrument(s) [26, 118], Iss106: Scarcity of coaching for utilizing groupware instruments [127], Iss107: Utilization of inadequate technique for eliciting requirements [Proposed], Iss108: Assumptions regarding instruments and Technologies are not fulfilled [proposed], Iss109: The instruments have security and scalability problems [Proposed].

**vi. Relationship among stakeholders' issues**, Iss110: Absence of steady relationship amongst the shareholders [93, 141], Iss111: Not passing on data, to identify or settle requirements related issues, to dispersed locations for a longer time span [92], Iss112: Rarity of casual interactions leads to fewer chances of establishing relations [100], Iss113: Utilization of various standards, by client and vendor, for documenting the requirements [26], Iss114: Creation of client or/and service provider teams on temporary base [26], Iss115: Disparate preferences of customer and vendor to collect and confirm requirements [26], Iss116: Less involvement of customer side during requirements engineering process [26, 33], Iss117: Team(s) from vendor side have misapprehensions regarding working practices of the client side [26], Iss118: Customer and vendor pursue contradictory approaches for requirements engineering [26], Iss119: Unsuccessfulness of vendor to meet due dates and satisfy the obligations regarding requirements [26], Iss120: Problems of deciding about requirements related deliverables [26], Iss121: Disagreement on choice of RE instruments [26], Iss122: Clients feel that executing requirements associated work from distant requirements is impassible [21], Iss123: Customer and service provider depend on verbal contract [105].

**vii. Requirements centric issues**, Iss124: Confirming requirements in case of all shareholders relying on the requirements collected or data acquired only from the accessible shareholders [129], Iss125: Requirements' descriptions are misunderstood [69, 142], Iss126: Inaccurate or wrong requirements [143], Iss127: Not creating the requirements founded on suitable business cases [144], Iss128: Gold-plated or additional requirements [144], Iss129: Uncompleted requirements [109, 137, 143], Iss130: No standards for documenting the requirements [145], Iss131: Inclusion of the requirements that are not within the scope [135], Iss132: Requirements are described/specified ambiguously [5, 21, 69, 109, 118, 146], Iss133: Not giving data or giving deliberately vague data about requirements [33, 102], Iss134: Non availability of the criterion to prioritize the requirements [118], Iss135: Requirements are altered again and again [5, 69, 109, 146], Iss136: Discrepancies in the requirements related documents [109], Iss137: Enlarging the requirements that causes scope slinking [5], Iss138: Requirements are elicited via

fragmentation, means various individuals finalize the requirements belonging to various system's fragments, that causes client displeasure [147], Iss139: Analysts are devoid of the tactics that are needed to address the requirements description issues in case of outsourced projects [105], Iss140: Just chosen shareholders are counseled to elicit the requirements that roots for prejudiced elicitation [148], Iss141: Actual end users and individuals who collaborate with the analysts are not same [121],

Iss142: Analysts are influenced to conceal certain data associated to requirements that grounds for compromises to elicit and describe the requirements [121], Issu143: Clients are uncertain regarding the software requirements [Proposed], Iss144: Analysts presume, in view of their expertise, that they are aware of the clients' requirements [Proposed], Iss145: Clients are intrigued by the services provided by various systems and desire that their system should provide similar facilities, however, actually they are not needed [Proposed], Issu146: Customers emphasis on including more requirements whereas cost and schedule have been settled [Proposed], Iss147: Absence of real clients currently [Proposed], Iss148: Employing a technique to elicit requirements but its appropriateness is not investigated [Proposed], Iss149: General approach to address the problem is incorrect [Proposed],

Iss150: Applying presumptions to confirm or conclude requirements [Proposed].

By conducting a Delphi questionnaire survey with SDO industry practitioners, the 150 issues have been ranked based on the 'frequency of occurrence'. For this purpose, a five-point Likert scale has been exploited: i. Almost always i.e. 90–100% time (5), ii. Frequently i.e. 60–89% time (4), iii. About half of the time i.e. 40–59% time (3), iv. Occasionally i.e. 10–39% time (2), and v. Rarely i.e. seldom or never (1). Every issue has been ranked from two perspectives: i. Category-wise that is within respective category of the issue, and ii. Overall that is with respect to all the other issues belonging to the respective category of the issues and all the other categories. Grounded on the 'frequency of occurrence' based ranking, study extracts 43 customarily arising issues of the SDO RE process. Out of the 43 customarily arising issues, six issues belong to 'communication' category and seven issues belong to 'knowledge management & awareness' category. The 'cultural diversities' category causes five issues. Five issues belong to 'management & coordination'. The 'Processes & tools' category has five issues, six issues are related to 'relationship among stakeholders' whereas nine issues are 'requirements centric'. The categories of the issues have also been ranked. The seven categories along with the corresponding ranks are: i. Communication (1), ii. Management & coordination (2),

iii. Knowledge management & awareness (3), iv. Requirements centric (4), v. Cultural diversities (5), vi. Processes & tools (6), vii. Relationship among stakeholders (7).

The study also highlights top 10 frequently occurring issues of the SDO RE process.

## 5.1 Top 10 customarily arising issues of the SDO RE process

The concept of highlighting the top 10 objects is quite prevalent. Sommerville & Sawyer indicate the top 10 practices for RE [84], Xindong & Kumar debate on the top 10 algorithms used for data mining [85] whereas J. M. Schopf reports the top 10 queries regarding grids [86]. T. Arnuphaptrairong notifies the top 10 listings related to risks involved in software development project [87]. Numerous studies focus on the top 10 risks concerning software projects [70, 88–90]. Thus, based on values given in Table 16, the top 10 customarily arising issues of the SDO RE process have been mentioned in Table 17. This can be observed from that out of the 11 customarily arising issues holding top 10 ranks, five issues are linked to communication, three issues are connected to knowledge management & awareness, and three issues are associated to management & coordination. The results illustrate that these aspects must be given topmost priority during the project management plan in the SDO context.

**Table 17. Top 10 customarily arising or common issues of the SDO RE process.**

| Sr. # | Issues and IDs | Means | Overall ranks | Categories |
|---|---|---|---|---|
| I₁ | Deferred replies [93, 99–100]. | 4.213592 | 1 | Communication |
| I₂ | Deficiency of casual correspondence amongst the shareholders [33, 91–93]. | 4.203883 | 2 | Communication |
| I₃ | Typically, there is non-recording of the promises that are done amid videoconferencing or discussions on the telephone, consequently such pledges cannot be alluded when needed [Proposed]. | 4.194175 | 3 | Communication |
| I₁₂ | Inadequate management of the modifications in requirements [69, 115]. | 4.165049 | 4 | Knowledge management and awareness |
| I₇ | Postponement in elucidations regarding requirements and finalizing decisions [94]. | 4.165049 | 4 | Management and coordination |
| I₁₃ | Unfamiliarity of the shareholders from existing/recent data regarding requirements [111]. | 4.165049 | 4 | Knowledge management and awareness |
| I₈ | Failure in performing RE associated assignment(s) as everyone believes this is obligation of another person [Proposed]. | 4.145631 | 7 | Management and coordination |
| I₄ | Deficiency of synchronized correspondence [96–97]. | 4.126214 | 8 | Communication |
| I₅ | Occasional and controlled correspondence amongst the shareholders [40]. | 4.116505 | 9 | Communication |
| I₉ | Improperly defined or vague obligations [118, 135]. | 4.106796 | 10 | Management and coordination |
| I₁₄ | Unfamiliarity with or not consulting all the origins of requirements [Proposed]. | 4.106796 | 10 | Knowledge management and awareness |

## 6. Limitations of the study

To conduct the study, the Convenience sampling method has been adopted and the participating SDO industry practitioners belong to only two countries.

To attain the objectives of the study, three questionnaire surveys have been conducting whereas two of these surveys involve very lengthy questionnaires. Keeping in view nature of the study, it was intended that same participants or at least participants from the same companies or organizations should participate in the surveys to complete the study. Time constraints were also there. In these circumstances, software development outsourcing practitioners or their representatives from various countries of the world were contacted. But the results were extremely disappointing as practitioners were busy or were not available at that particular time. Therefore, Convenience sampling method was adopted. Through the Convenience sampling, those practitioners were included in the study who were willing to participate in the study upon our personal request or because of any academic or industrial reference. At the same time, for sake of quality, it was ensured that:

i. All the participants belong to the companies or organization which deal with software development outsourcing.

ii. All the participants have at least five years' experience of software development outsourcing related professional job.

iii. Participants belong to various professional categories like project manager, quality assurance manager, software engineer, team lead, requirements engineer, analyst, programmer etc.

iv. Participants have vast experience of dealing with a wide range of projects like embedded systems, telecommunication systems, business systems, e-commerce systems, multimedia applications, web-based systems, safety critical systems, accounting and finance systems, billing services systems.

v. Most of the respondents' companies or organizations are certified. Some of the companies or organizations are non-certified.

vi. Respondents' companies or organizations vary in size from small to medium and large.

vii. The number of respondents in the case of each survey is reasonable (more than 100).

viii. Respondents' companies or organizations run the business at national, regional and international level. Therefore, the participants have the experience of dealing with the professionals belonging to various backgrounds and cultures. Based on their exposure, the participants have skills of addressing communication, knowledge management and coordination issues.

Keeping in view all these facts, the sample(s) can be safely considered as representative of the large population.

## 7. Conclusion and future directions

Taking into account the anticipated benefits of Software Development Outsourcing (SDO) and reasons for the SDO failure, this study explores and highlights the commonly arising issues of the Requirements Engineering (RE) process in the case of SDO. Many a time RE process issues jeopardize SDO projects and eventually such project are failed. To evade the 'fire fighting' approach for tackling the SDO RE process issues and for successfully addressing such issues to attain the SDO benefits, the issue must be contemplated beforehand based on 'frequency of occurrence'.

This study explores the issues of the RE process for SDO. The issues belong to various categories. Thus, firstly this study identifies seven categories of the RE process issues for SDO that are:

i. Communication, ii. Knowledge management and awareness, iii. Cultural diversities,

iv. Management and coordination, v. Processes and tools, vi. Relationship among stakeholders, and vii. Requirements centric (RQ1).

To **devi**se a pragmatic proactive strategy for addressing the SDO RE process issues, the commonly occurring SDO RE process issues must be identified. Therefore, 43 customarily arising SDO RE process issues have been excavated from the list of total 150 issues (129 issues from literature and 21 from SDO industry). Out of the 43 issues, six issues belong to 'communication' category and seven issues belong to 'knowledge management and awareness' category. Similarly, 'cultural diversities' category causes five issues. Furthermore, five issues belong to 'management and coordination'. The 'processes and tools' category has five issues, six issues are related to 'relationship among stakeholders' whereas nine issues are from 'requirements centric' category (RQ2). Ranking of the issues is also essential for dealing with the issues. Therefore, the ranks of the issues have been ascertained hinging on the 'frequency of occurrence' of the issues by incorporating a five-point Likert scale: i. Almost always i.e. 90 to 100% time (5), ii. Frequently i.e. 60 to 89% time (4), iii. About half of the time i.e. 40 to 59% time (3), iv. Occasionally i.e. 10 to 39% time (2), and v. Rarely i.e. seldom or never (1). The two ranks have been associated with each issue: i. Category-wise rank, and ii. Overall rank. The Category-wise rank provides the rank of an issue with respect to all the other issues within the respective category of the issue (RQ3.1) whereas the Overall rank provides the rank of an issue with respect to all the other issues belonging to all the seven categories (RQ3.2). The seven categories of the frequently arising issues have also been ranked. The seven categories along with the respective ranks are:

i. Communication (1), ii. Management and coordination (2),

iii. Knowledge management and awareness (3), iv. Requirements centric (4),

v. Cultural diversities (5), vi. Processes and tools (6), vii. Relationship among stakeholders (7) (RQ3.3).

The study also presents the top 10 issues of the SDO RE process. The identification of the commonly occurring SDO RE process issues and the ranking of the issues, helps executives and managers in planning a proactive strategy for dealing with the SDO RE process issues and hence to achieve prophesied benefits of SDO.

As the future work, the plan is to:

i. Identify the root-causes for the commonly occurring issues of the RE process in the case of SDO, for this purpose Root Cause Analysis would be performed.

ii. Purpose a model for addressing the issues of SDO RE process.

## Supporting information

**S1 File. Literature assessment.** Literature assessment details.
(ODT)

**S2 File. Survey participants.** Survey participants' details and recruitment.
(ODT)

**S1 Appendix. Consolidated list of RE process issues in case of software development outsourcing.**
(DOCX)

**S2 Appendix. Average frequency & standard deviation, for each issue, calculated after 2$^{nd}$ & 3$^{rd}$ Delphi rounds.**
(DOCX)

## Author Contributions

**Data curation:** Javed Iqbal, Fazal-e-Amin, Adnan Akhunzada, Muhammad Shoaib.

**Formal analysis:** Javed Iqbal, Rodina B. Ahmad, Muzafar Khan, Fazal-e-Amin, Sultan Alyahya, Adnan Akhunzada, Muhammad Shoaib.

**Funding acquisition:** Fazal-e-Amin, Sultan Alyahya, Muhammad Shoaib.

**Investigation:** Javed Iqbal, Muzafar Khan, Sultan Alyahya.

**Methodology:** Javed Iqbal, Mohd Hairul Nizam Nasir.

**Project administration:** Rodina B. Ahmad, Mohd Hairul Nizam Nasir.

**Resources:** Rodina B. Ahmad, Mohd Hairul Nizam Nasir.

**Supervision:** Rodina B. Ahmad.

**Validation:** Javed Iqbal.

**Writing – original draft:** Javed Iqbal, Muzafar Khan.

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
