## [Decision Letter · Decision Letter 0]

30 Dec 2019

PONE-D-19-29952

Software development outsourcing failure: Identification, categorization and ranking of the customarily occurring requirements engineering issues

PLOS ONE

Dear Dr. Iqbal,

Thank you for submitting your manuscript to PLOS ONE. After careful consideration, we feel that it has merit but does not fully meet PLOS ONE’s publication criteria as it currently stands. Therefore, we invite you to submit a revised version of the manuscript that addresses the points raised during the review process.

I recommend that it should be revised taking into account the changes requested by the first reviewer. I would like to give you a chance to revise your manuscript. However, the manuscript will only be reviewed by the Academic Editor in the next round.

We would appreciate receiving your revised manuscript by Feb 13 2020 11:59PM. To enhance the reproducibility of your results, we recommend that if applicable you deposit your laboratory protocols in protocols.io, where a protocol can be assigned its own identifier (DOI) such that it can be cited independently in the future. For instructions see: http://journals.plos.org/plosone/s/submission-guidelines#loc-laboratory-protocols

We look forward to receiving your revised manuscript.

Kind regards,

Baogui Xin, Ph.D.

Academic Editor

PLOS ONE

Journal Requirements:

3. Thank you for including your ethics statement: 

"The research work for this study, being part of PhD work, has been approved by the committee for candidature défense. The questionnaire surveys are only human related subject of this study. Before conducting the surveys, the verbal consent has been obtained from the potential participants or from their respective organizations. No personal data has been presented or analyzed in any form in this study. The responses have been presented in an accumulative manner. In this way, privacy and anonymity of the individuals and organizations have been fully protected."

Reviewers' comments:

Reviewer's Responses to Questions

**Comments to the Author**

1. Is the manuscript technically sound, and do the data support the conclusions?

Reviewer #1: Yes

Reviewer #2: Yes

2. Has the statistical analysis been performed appropriately and rigorously? 

Reviewer #1: Yes

Reviewer #2: Yes

3. Have the authors made all data underlying the findings in their manuscript fully available?

Reviewer #1: Yes

Reviewer #2: Yes

4. Is the manuscript presented in an intelligible fashion and written in standard English?

Reviewer #1: Yes

Reviewer #2: Yes

5. Review Comments to the Author

Reviewer #1: The paper seems technically sound and very intelligently written and hence is a good addition in the software engg domain. Following minor improvements are suggested to further improve the paper.

1. The title of the paper is too long and sounds ambiguous. The authors may consider to re-structure the title.

2. Throughout the text the author has not taken care of the capital and small letters.

3. Page 3, Line 51, "The RE related errors occur commonly during 62 SDLC [25]. According to an industrial survey, such errors are 48%" the statement needs to be justified. The context is not clear.

4. Page 15, Table 1. the table should also mention the % of finally accepted papers as compared to initially identified papers.

5. Page 12, Table 2, the identified issues and shortfalls need to be justified.

6. Before conclusion section, the author should have a detailed discussion section.

7. Limitation and future work section needs to be added.

Reviewer #2: Author has performed an extensive and detailed study. He resulted with great outcomes.

Authors have done excellent job for Software development outsourcing failure: Identification, categorization and ranking of

the customarily occurring requirements engineering issues.

A detail delphi process is adopted on extensive data set and a good quality of evidence were collected from literature too to support their claim.

6. PLOS authors have the option to publish the peer review history of their article (what does this mean?). If published, this will include your full peer review and any attached files.

Reviewer #1: Yes: Dr Basit Shahzad

Reviewer #2: Yes: Kinza Mehr Awan

---

## [Author Response · Author response to Decision Letter 0]

12 Feb 2020

REVIEWER 1:

1: The title of the paper is too long and sounds ambiguous. The authors may consider to re-structure the title.

Response:The title has been changed. 

2: Throughout the text the author has not taken care of the capital and small letters.

Response:The paper has been reviewed thoroughly and reviewer’s concern has been addressed. The changes have been highlighted.

3: Page 3, Line 51, "The RE related errors occur commonly during 62 SDLC [25]. According to an industrial survey, such errors are 48%" the statement needs to be justified. The context is not clear.

Response:More information have been added to make the context clear.

4: Page 15, Table 1. the table should also mention the % of finally accepted papers as compared to initially identified papers.

Response:A new column has been introduced in Table 1 to mention required percentage.

5: Page 12, Table 2, the identified issues and shortfalls need to be justified.

Response:Done as recommended.

6: Before conclusion section, the author should have a detailed discussion section.

Response:Discussion section has been added.

7: Limitation and future work section needs to be added.

Response:Limitations and future work have been added.

REVIEWER 2:

Author has performed an extensive and detailed study. He resulted with great outcomes.

Authors have done excellent job for Software development outsourcing failure: Identification, categorization and ranking of

the customarily occurring requirements engineering issues.

A detail Delphi process is adopted on extensive data set and a good quality of evidence were collected from literature too to support their claim. 

Response: We appreciate the comment.

---

## [Editor Report · Decision Letter 1]

14 Feb 2020

Requirements engineering issues causing software development outsourcing failure

PONE-D-19-29952R1

Dear Dr. Iqbal,

We are pleased to inform you that your manuscript has been judged scientifically suitable for publication and will be formally accepted for publication once it complies with all outstanding technical requirements.

With kind regards,

Baogui Xin, Ph.D.

Academic Editor

PLOS ONE
---

## [Editor Report · Acceptance letter]

4 Mar 2020

PONE-D-19-29952R1 

Requirements engineering issues causing software development outsourcing failure 

Dear Dr. Iqbal:

I am pleased to inform you that your manuscript has been deemed suitable for publication in PLOS ONE. Congratulations! Your manuscript is now with our production department. 

With kind regards,

on behalf of

Prof. Baogui Xin 

Academic Editor

PLOS ONE